# Deep Learning for Optical Sensor Applications: A Review

**DOI:** 10.3390/s23146486

**Published:** 2023-07-18

**Authors:** Nagi H. Al-Ashwal, Khaled A. M. Al Soufy, Mohga E. Hamza, Mohamed A. Swillam

**Affiliations:** 1Department of Physics, The American University in Cairo, New Cairo 11835, Egypt; nagi.alashwal@ibbuniv.edu.ye (N.H.A.-A.); kalsoufi@gmail.com (K.A.M.A.S.); mohga.essam@aucegypt.edu (M.E.H.); 2Department of Electrical Engineering, Ibb University, Ibb City 00967, Yemen

**Keywords:** deep learning, optical sensors, deep neural network, convolutional neural network, autoencoders

## Abstract

Over the past decade, deep learning (DL) has been applied in a large number of optical sensors applications. DL algorithms can improve the accuracy and reduce the noise level in optical sensors. Optical sensors are considered as a promising technology for modern intelligent sensing platforms. These sensors are widely used in process monitoring, quality prediction, pollution, defence, security, and many other applications. However, they suffer major challenges such as the large generated datasets and low processing speeds for these data, including the high cost of these sensors. These challenges can be mitigated by integrating DL systems with optical sensor technologies. This paper presents recent studies integrating DL algorithms with optical sensor applications. This paper also highlights several directions for DL algorithms that promise a considerable impact on use for optical sensor applications. Moreover, this study provides new directions for the future development of related research.

## 1. Introduction

Modern digital development is the combination of sensors (hardware) and artificial intelligence (AI) (software) to perform intelligent tasks, now a key component in machine learning (ML). ML, a branch of AI, has a significant impact on optical sensors. This new model takes a data-driven approach without focusing on the underlying physics of the design. It also brings forth new advancements to conventional design tools and opens up numerous opportunities. Sensing has a significant impact on a broad range of scientific and engineering problems [1,2,3]. There are many types of sensors, one of them being optical sensors. Optical sensors have many useful features, including their light-weight, low-cost, small size, flexible deployment, and ability to operate at high pressures [4], high/low temperatures [5,6], and in high electromagnetic fields [7] without a reduction in their performance. Due to these advantages, optical sensors have been used in many applications such as intrusion detection [8], the monitoring of railways and general transport [9], pipelines [10], and bridge structures [11]. They also are used in the detection and localization of seismic events [12], human event recognition [13], healthy tasks [14], building structure [15], and landslide detection [16].

The use of multiple sensors generates huge raw datasets, causing serious challenges in processing and managing the data. Furthermore, conventional processing techniques in traditional sensing devices are not suitable for labelling, processing, or analysing the data [17]. Moreover, the collected data require a long time to be processed. The cost is another problem, where some applications require the deployment of many sensors. Deep learning (DL), a branch of ML, is incorporated with optical sensors to solve the aforementioned challenges [18,19,20,21]. Deep neural networks (DNNs) are a modern and promising technology that can be used with various optical sensors applications. The main advantage of DNNs is their ability to dynamically extract features from the collected raw data with high accuracy, often outperforming the capability of humans [22,23].

In the state-of-the-art research in this field, some previous survey papers have reviewed the use of DL in specific applications for optical sensors. As an example, the authors in [24] presented an extensive review of the recent advances in the estimation of multiphase fluid flows. The distributed optical fibre sensors and their working mechanism were addressed. The article provided some recent works which were used to characterize multiphase fluid flows in the production optimization of the oil and gas industry. It also included traditional methods, such as estimation of the sound speed and Joule–Thomson coefficient, in addition to data-driven ML techniques such as CNN, ensemble Kalman filter (EnKF) and support vector machine (SVM) algorithms. Some related papers that used CNN and ANN models to perform flow regime classification and multiphase estimation are mentioned in [10,25]. The LSTM algorithm was adopted by other related papers to estimate fluid flow rate [26,27,28,29]. Another survey includes ref. [30], in which the authors presented the latest advancements in pattern recognition models utilized in distributed optical fibre vibration sensing systems (DOVS). The main issues presented were feature extraction, the structure of the model, and the model performance. Some applications were introduced, including railway safety monitoring, perimeter security and pipeline monitoring. The authors also provided the pattern recognition prospects for DOVS in addition to some related references which realized the pattern recognition of vibration signals using ML and DL. In [15], the authors reviewed the current state of optical sensors and the application of DL for the structural health monitoring of civil infrastructures. The review considered the past five years and found that optical fibre sensors were applied to the measurement of concrete properties, leakage monitoring, corrosion, and fatigue responses.

The objective of this work is to review DL for optical sensor applications. This review comprehensively covers all published optical sensor types that have been utilized in conjunction with DL techniques. DL can benefit optical sensors in many directions such as processing huge datasets, pre-processing noisy data, automating feature extraction, predicting the results with high accuracy and reliability, and reducing the number of optical sensors required for the deployment in any system. To the best of our knowledge, this is the first study that discusses DL applications for optical sensors.

This paper is organized as follows. In the Section 2, we introduce the operating principles of optical sensors. In the Section 3, we present a brief discussion on DL. A survey of the application of DL on optical sensors is presented in the Section 4. A discussion and future perspectives are given in the Section 5.

## 2. Optical Sensor Technologies

Since the 1900s, plasmonic sensors have been widely used in many areas for numerous applications [31,32,33,34,35]. Since then, they have been utilized in a diverse range of fields, including biology and medical diagnosis [36,37,38,39,40], chemistry [41], food safety [42,43], and environmental monitoring and evaluation [44,45,46,47,48]. In addition, plasmonic sensors are being used in negative refractive index materials [49,50,51], optical meta-surfaces [52,53,54], and integrated circuits [55,56,57]. Consequently, the effects caused by the surface plasmon resonance (SPR) or localized surface plasmon resonance (LSPR) have been proven to have an astounding sensitivity needed in those applications [58]. To design plasmonic sensors, the appropriate selection of the operating wavelength in addition to the type and thickness of the metal film to be used is required to achieve optimum sensitivity [59]. If the sensors are used in the visible range and the near-infrared range of the electromagnet spectrum, the most typical metals used are gold, copper, silver, and aluminium since they have the sharpest resonance compared to other metals [60]. Among these metals, gold is mostly preferred since it is the most chemically stable when exposed to the atmosphere. However, gold does not show SPR when the wavelength used is less than 0.5 μm [61,62].

The SPR functions when photons from the incident light directed onto a metal surface layer excite the conductive electrons on its surface at a specific angle to undergo collective oscillations and then be propagated parallel to the surface [63]. This occurs since the surface of the SPR is highly sensitive due to the SPR-generated evanescent field, occurring at the interface between the metal and the dielectric while undergoing total internal reflection. The point of interface is considered the strongest part at which the evanescent field happens because of the resonance coupling between the incident rays and the surface plasmon wave [64]. As the evanescent field infuses into the dielectric media and the metal film, it decreases exponentially. The greatest decay of the field makes the SPR sensors significantly sensitive to the thickness of the material and the refractive index alterations of the dielectric film affixed to the metal-based surface [65,66,67,68,69]. The binding occurring at the surface of the metal and the thickness of the dielectric film affect the signal measured by the plasmonic sensor [70]. This happens since the resonance of the surface plasmon wave shifts with any changes in the thickness of the material, observed when the SPR curve shifts [64]. There is a linear relationship between the SPR signal and the dielectric film thickness and the refractive index, which facilitates SPR spectroscopy of the interaction happening to be quantitatively analysed. Hence, studying the SPR signal as a function of time explains the binding kinetics and interactions occurring at the plasmonic sensors to be measured in real-time [70].

The measurement of the reflective index changes along with the binding of the sample for recognition of the immobilized molecules on the SPR sensor, as shown in Figure 1. Hence, the structures’ size, shape, and composition along with the dielectric properties of the neighbouring environment, utterly determines the intensity and position of the SPR, which are key components in creating an optical sensor [71,72]. Hence, any minor adjustment to the reflective index of the sensing medium would alter the SPR occurrence which is used for detecting the analyte or chemical [73,74]. The numerous variables involved in the analysis of the SPR certify high sensitivity, making it highly important to be utilized in various applications [69]. Another sensor is nanoparticle-based plasmonic biosensors, which have high sensitivity and low LOD so they have been employed in pathogen detection, allowing for the detection of various diseases due to the wide spectrum of antibody binding. They are especially crucial for POCT because, when employed as biosensors, they are non-intrusive, quick, and accurate. The sensitivity of plasmonic biosensors can be further increased by adding metamaterials, enabling the biosensor to be reliable and reproducible. Recently, some researchers have been focusing on improving biosensors so that they might be created as a lab-on-a-chip diagnostic tool, making them ubiquitous. Additionally, it is being used by other researchers to look for airborne illnesses in the environment. Metamaterial-based plasmonic biosensors would enable highly accurate and rapid detection of pathogens that could improve human well-being and shield humanity from any pandemic in the future, regardless of their physical or airborne modes of transmission [36].

## 3. Deep Neural Networks Overview

DL, a subset of ML, has a great advantage due to its ability to automatically learn representative features from input data, identified as “feature learning”. DL has shown outstanding success due to having large datasets, partaking ongoing advances in computing power, and having an enduring opportunity in algorithm improvements. DL uses several DNNs to carry out intricate computations on enormous volumes of structured and unstructured data. There are three types of learning in DNNs. The first type is a supervised algorithm which works with labelled data. At this point, the model is trained to reduce the cost function which reflects the difference between the model’s predictions and the actual values.

CNN [75] and LSTM [76,77] are examples of this type. The second type is a semi-supervised algorithm where a small part of the sample has the annotations necessary to train the model. This form of algorithm constructs a self-learning strategy where it generates its own annotations [78]. Examples of this type include generative adversarial networks (GAN) [79] and deep reinforcement learning (DRL) [80]. The third type of DNN is an unsupervised algorithm where the model finds the structure or relationship between the input data without labels or annotations. Restricted Boltzmann machines (RBM) [81] and AE [82] are examples of this type of DNN and they perform dimensionality reduction functions or clustering.

In optical sensor applications, the most commonly adopted learning architectures are CNN, AEs, and multiple layer perceptron (MLP) [83].

### 3.1. Convolutional Neural Network (CNN)

The CNN has three types of layers, including the convolutional layer, which works as a feature extractor from the input image, the pooling layer, which reduces the dimensionality of features maps [84], and the fully connected layer, which is located near the output layer. A SoftMax classifier is usually used as the final output layer as shown in Figure 2. In synchronization, these layers enable a learning scheme that links the map features to predict the output and minimize the cost function. By employing a shared kernel in the convolution operation, DL models are able to learn space-invariant features. Furthermore, in comparison with fully connected neural networks, CNNs are good at capturing local dependencies. As for LSTMs, they are employed to make time-series predictions as they resolve the issue of the vanishing gradient which arises in conventional recurrent neural networks (RNNs) [76].

### 3.2. Autoencoders (AE)

An AE is a neural network designed with the objective of learning a representation that closely mirrors the input data when presented as an output [82,85]. As shown in Figure 3, the AE consists of two components, the encoder and the decoder. The input and output layers have the same number of neurons while all the layers are interconnected. However, the network incorporates a bottleneck to encourage the learning of the essential features only. To create a bottleneck effect in the AE, the number of nodes in the connecting layer, located between the encoder and the decoder parts, are reduced in comparison to nodes in the input layer. Similar to other neural networks, the training process of the AE involves learning the weights and biases of the network by minimizing the loss functions. This ensues as the encoder component learns a compact representation of the input data, while the decoder component reconstructs the original input from the learned representation provided by the encoder. The process of learning and reconstruction in the AE has been used in a range of applications, including anomaly detection. By leveraging the learned representation and reconstruction capability, the AE can effectively identify anomalies or deviations from the normal patterns in the input data. This enables the AE to serve as a valuable tool for detecting and flagging unusual or anomalous occurrences in various domains.

### 3.3. Multilayer Perceptron (MLP)

The multilayer perceptron (MLP) is a complement of the feedforward artificial neural network. It comprises three layers, including the input layer, the output layer, and the hidden layer, as shown in Figure 4. The input layer is responsible for receiving the input signal, which needs to be processed. The output layer is responsible for performing the desired task, such as prediction or classification. The true computational engine of the MLP lies within an arbitrary number of hidden layers, positioned between the input and the output layers. These hidden layers carry out the complex computations and transformations that enable the MLP to learn and extract meaningful patterns from the input data. In an MLP, data follows a similar flow to a feedforward network, progressing from the input layer to the output layer in a forward direction. The neurons within the MLP are trained via the usage of the backpropagation learning algorithm. MLPs are specifically designed to approximate any continuous function and are capable of solving problems that are not linearly separable. Some of the most significant applications of MLP include pattern classification, recognition, prediction, and approximation.

## 4. DL Applications for Optical Sensors

DL development is a highly iterative and empirical process. It can be implemented in three steps, including choosing the initial weights and hyperparameter values, coding, and experimenting. These steps are interconnected through an interactive process.

For optical sensor applications, the first step is to figure out the current problem with those applications such as the processing of vast data, noisy data, missing data, and the delay in data processing. Then, an innovative idea needs to be formed to integrate DL into the optical sensor to solve these complications. The next step is to code the proposed solution using related and modern frameworks or toolkits, such as TensorFlow, Keras, PyTorch, CNTK, etc. After this, training and evaluating the model needs to be conducted by having raw data gathered, pre-processed, and fed into the proposed DL model. Based on the results, the developer should refine the proposed model cyclically and apply any required amendments in the model to obtain better accuracy. The overall view of these steps is depicted in Figure 5. Optical sensor applications based on DL techniques are surveyed here to provide interested researchers and readers with the elementary knowledge for developing high-performance optical sensors. Furthermore, this work introduces and discusses the most recent, related applications, which is mainly focusing on some factors and issues such as motivation, strategy, and effectiveness. The content of the survey is further expanded according to the used DL model to which each work belongs.

### 4.1. CNN-Based Applications

CNN-based applications are attracting interest across a variety of domains, including optical sensor applications. In this section, some of the recent works that have been applying this model for optical sensors will be briefly presented

In [86], a CNN model was developed to comprehend an optical fibre curvature sensor. A large number of specklegrams have been automatically detected from the facet of multimode fibres (MMFs) in the experiments. The detected specklegrams were pre-processed and fed to the model for training, validation, and testing. The dataset was collected in the form of a light beam by designing an automated detecting experimental setup as shown in Figure 6. The light beam was detected by a CCD camera with a resolution of 1280×960 and a pixel size of 3.75 × 3.75 μm${}^{2}$. As shown in Figure 7, the architecture of VGG-Nets was adopted to build the CCN. The mean squared error (MSE) was then used as the loss function. The predicted accuracy of the proposed CNN was 94.7% of specklegrams with an error of curvature prediction within 0.3 m${}^{-1}$. However, the learning-based scheme reported has the capability to only predict a solitary parameter and does not fully utilize the potential of DL.

In [9], the authors proposed semi-supervised DL to detect a track. An experimental setup was created using a portion of a high-speed railway track and installing a distributed optical fibre acoustic system (DAS). In the proposed model, an image recognition model with a specific pre-processed dataset and an acquisitive algorithm for selecting hyperparameters was used.

The considered events supposed to be recognized in this model are shown in Table 1.

In addition, the hyperparameters were selected based on an acquisitive algorithm. The obtained dataset after the augmentation process is shown in Table 2.

Four structural hyperparameters were used in this work as shown in Table 3. The obtained accuracy of the proposed model was 97.91%. However, it is important to highlight that the traditional methods have proven to execute improved spatial accuracy. Some other related works can be found in [87,88,89,90,91,92,93,94].

In [95], a distributed optical fibre sensor using a hybrid Michelson–Sagnac interferometer was proposed. The motivation of the proposed model was to solve the complications of the incapability of the conventional hybrid structure to locate in the near and flawed frequency response. The proposed model utilized basic mathematical operations and a 3×3 optical coupler to obtain two phase signals with a time difference that can be used for both location and pattern recognition. The received phase signals were converted into 2D images. These images were used as a dataset and fed into the CNN to obtain the required pattern recognition. The dataset contained 5488 images with six categories, and the size of each image was 5000×5000 in .jpg format. The description of the dataset is shown in Table 4. The structural diagram of the used CNN is shown in Figure 8. The accuracy of the proposed model was 97.83%. However, the sensing structure employed is relatively simple and does not consider factors such as the influence of backward scattered light.

In [96], a DL model was proposed to extract time–frequency sequence correlation from signals and spectrograms to improve the robustness of the recognition system. The authors designed a targeted time attention model (TAM) to extract features in the time–frequency domain. The architecture of the TAM model comprises two stages, namely the convolution stage for extracting features and the time attention stage for reconstruction. The process of data streaming, domain transformation and features extraction to output is shown in Figure 9. The knocking event is taken as an example. The convolution stage is used to extract characteristic features. Here, the convolutional filter established a local connection in the convolution and shared the weights between receiving domains. The pooling layers emphasized the shift-invariance feature. In addition, a usual CNN model was used as the backbone. As shown in Figure 9, in the left stage, information was extracted from the spectrogram and transformed into a feature map (1×128×200), where 1 represents the number of input channels (the grey image has one channel), while 128 and 200 represent the height and width of the input, respectively. The authors collected and labelled a large dataset of vibration scenes including 48,000 data points with eight vibration types. The experimental results indicated that this approach significantly improved the accuracy with minimal additional computational cost when compared to the related experiments [97,98]. The time attention stage was designed for the reconstruction of the features in which TAM was used to serve two purposes. The first purpose was to extract the sequence correlation by a cyclic element. The second purpose was to assign the weight matrices for the attention mechanism. F1 and F2 were unique in their emphasis on investigating the "where" and "what" features of time. An F-OTDR system was constructed to classify and recognize vibration signals. The F-OTDR system contained a sensing system and a producing system. This study was verified using a vibration dataset including eight different scenarios which were collected by an F-OTDR system. The achieved classification had an accuracy of 96.02%. However, this method did not only complicate the data processing procedure, but it also had the potential to result in the loss of information during the data processing phase.

In [99], a real-time action recognition model was proposed for long-distance oil–gas PSEW systems using a scattered distributed optical fibre sensor. They used two methods to calculate two complementary features, a peak and an energy feature, which described the signals. Based on the calculated features, a deep learning network (DLN) was built for a new action recognition. This DLN could effectively describe the situation of long-distance oil–gas PSEW systems. The collected datasets were 494 GB with several types of noise at the China National Petroleum Corporation pipeline. The collected signal involved four types of events containing background noise, mechanical excavation, manual excavation, and vehicle driving. As shown in Figure 10, the architecture of the proposed model consisted of two parts. The first part dealt with the peak, while the second part dealt with the energy. Each part consisted of many layers, including ConvD1, batch normalization, maxpool, dropout, Bi-LSTM, and a fully connected layer. Any damage event could be allocated and identified with accuracies of 99.26 and 97.20% at 500 and 100 Hz, respectively. Nonetheless, all the aforementioned methods consider an acquisition sample as a singular vibration event. However, for dynamic time-series identification tasks, the ratio of valid data within a sample relevant to the overall data was not constant. This means that the position of the label in relation to the valid portion of the input sequence remained uncertain. Further related research can be reviewed in [100,101,102].

In [103], the authors presented the application of signal processing and ML algorithms to detect events using signals generated based on DAS along a pipeline. ML and DL approaches were implemented and combined for event detection as shown in Figure 11. A novel method to efficiently generate training dataset was developed. Excavator and none-excavator events were considered.

The sensor signals were converted into a grey image to recognize the events depending on the proposed DL model. The proposed model was evaluated in real-time deployment within three months in a suburban location as shown in Figure 12.

The results showed that DL is the most promising approach due to its advantages over ML as shown in Table 5. However, the proposed model only differentiated between two events, namely ‘excavator’ and ‘non-excavator’, while there are multiple distinct events. Additionally, the system was evaluated in a real-time arrangement for a duration of three months in a suburban area. Yet, for further validation and verification, it is crucial to conduct tests in different areas and over an extended period of time.

In [104], an improved WaveNet was applied to recognize manufactured threatening events using distributed optical fibre vibration sensing (DVS). The improved WaveNet is called SE-WaveNet (squeeze and excitation WaveNet). WaveNet is a 1D CNN (1D-CNN) model. As a deep 1D-CNN, it can quickly achieve training and testing while also boasts a large receptive field that enables it to retain complete information from 1D time-series data. The structure of the SE functions in synchronization with the residual block of WaveNet in order to recognize 2D signals. The SE structure functions using an attention mechanism, which allows the model to focus on channel features to obtain more information. It can also suppress insignificant channel features. The structure of the proposed model is shown in Figure 13. The input of the SE-WaveNet is an n × m matrix, synthesized from the n-points of spatial signals and the m-groups of time signals. The used dataset is shown in Table 6. The results showed that the SE-WaveNet accuracy can reach approximately 97.73%. However, it is important to note that the model employed in this study was only assessed based on a limited number of events, and further testing is necessary to evaluate its performance in more complex events, particularly in engineering-relevant applications. Additionally, further research is needed to validate the effectiveness of the SE-WaveNet in practical, real-world settings.

In [14], a CNN and an extreme learning machine (ELM) were applied to discriminate between ballistocardiogram (BCG) and non-BCG signals. CNNs were used to extract relevant features. As for ELM, it is a feedforward neural network that takes the features extracted from CNN as the input and provides the category matrix as an output [105]. The architecture of the proposed CNN-ELM and the proposed CNN are shown in Figure 14 and Table 7, respectively.

BCG signals were obtained with a micro-bend fibre optical sensor based on IoT, taken from ten patients diagnosed with obstructive sleep apnoea and submitted for drug-induced sleep endoscopy. To balance the BCG (ballistocardiogram) and non-BCG signal samples, three techniques were employed: undersampling, oversampling, and generative adversarial networks (GANs). The performance of the system was evaluated using ten-fold cross-validation. Using GANs to balance the data, the CNN-ELM approach yielded the best results with an average accuracy of 94%, a precision of 90%, a recall of 98%, and an F-score of 94%, as shown in Table 8. Inspired by [106], the architecture of the used model is presented in Figure 15, showing balanced BCG and non-BCG chunks. Other relevant works were presented in [107,108].

In [11], the efficiency and accuracy enhancements of the bridge structure damage detection were addressed by monitoring the deflection of the bridge using a fibre optic gyroscope. A DL algorithm was then applied to detect any structural damage. A supervised learning model using CNN to perform structural damage detection was proposed. It contained eleven hidden layers constructed to automatically identify and classify bridge damage. Furthermore, the Adam optimization method was considered and the used hyperparameters are listed in Table 9. The obtained accuracy of the proposed model was 96.9% which was better than the random forest (RF), which was 81.6%, the SVM which was 79.9%, the k-nearest neighbor (KNN) which was 77.7%, and the decision trees (DT). Following the same path, comparable work was performed in [109,110].

The authors in [111] proposed an intrusion pattern recognition model based on a combination of Gramian angular field (GAF) and CNN, which possessed both high speed and accuracy rate in recognition. They used the GAF algorithm for mapping 1D vibration-sensing signals into 2D images with more distinguishing features. The GAF algorithm retained and highlighted the distinguishing differences of the intrusion signals. This was beneficial for CNN to detect intrusion events with more subtle characteristic variation differences. A CNN-based framework was used for processing vibration-sensing signals as input images. According to the experimental results, the average accuracy rate for recognizing the three natural intrusion events, light rain, wind blowing, and heavy rain, and the three human intrusion events, impacting, knocking, and slapping, on the fence was found to be 97.67%. With a response time of 0.58 seconds, the system satisfied the real-time monitoring requirements. By considering both accuracy and speed, this model achieved automated recognition of intrusion events. However, the application of complex pre-processing and denoising techniques to the original signal presented a challenge for intrusion recognition systems when it came to effectively addressing emergency response scenarios. Relevant work following a similar pattern was presented in [112].

A bending recognition model using the analysis of MMF specklegrams with diameters of 105 and 200 µm was proposed and assessed in [113]. The proposed model utilized a DL-based image recognition algorithm. The specklegrams detected from the facet of the MMF were subjected to various bendings and then utilized as input data. Figure 16 shows the used experimental setup to collect and detect fibre specklegrams.

The architecture of the model was based on VGG-Nets as shown in Figure 17.

The obtained accuracy of the proposed model for two multimode fibres is shown in Table 10.

More related research can be found in [113,114,115,116,117,118,119,120,121].

The authors in [122] used a CNN to demonstrate the capability for the identification of specific species of pollen from backscattered light. Thirty-core optical fibre was used to collect the backscattered light. The input data provided to the CNN was from camera images which were further divided into two sets, distance prediction and particle identification. In the first type, the total number of collected images was 1500, by which 90% of them were used as a training set and 10% were used as a validation set of the CNN. In the second type, 2200 images were collected and 90% of them were used as a training set and 10% were used as a validation set. The training procedure of the proposed model is depicted in Figure 18. The second version of ResNet-18 ([123,124]) was used to propose the required model with batch normalization [125] with a mini-batch size of 32 and a momentum of 0.95. The output was a single regression (single output). The neural network, trained to identify pollen grain types, achieved a real-time detection accuracy of approximately 97%. The developed system can be used in environments where transmission imaging is not possible or suitable.

In [13], a DL-based distributed optical fibre-sensing system was proposed for event recognition. A spatio-temporal data matrix from an F-OTDR system was used as the input data served to the CNN. The proposed method had advantageous characteristics, such as grey-scale image transformation and bandpass filtering, which were needed for pre-processing and classification instead of the usual complex data processing, small size, and high training speed, and excessive requirements for classification accuracy. The developed system was applied to recognize five distinct events involving background, jumping, walking, digging with a shovel, and striking with a shovel. The collected data were split into two types as shown in Table 11. The combined dataset for the five events consisted of 5644 instances.

Some common CNNs were examined, and the results are shown in Table 12.

The considered training parameters for all CNNs were the same. The total training steps were 50,000, the learning rate was 0.01, and the adopted optimizer was the root mean square prop (RMSProp) [126]. This work concluded that the VGGNet and GoogLeNet obtained better classification accuracy (greater than 95%) and GoogLeNet was selected to be the basic CNN structure due its model size. For further improvement of the model, Inception-v3 of GoogLeNet was used. Table 13 shows the classification accuracy achieved for the five events. The authors also optimized the network by tuning the size of some layers of the model. Table 14 shows the comparison between the optimized model and Inception-v3. However, it is important to note that this study trained the network using relatively small datasets consisting of only 4000 samples. Moreover, traditional data augmentation strategies employed in image processing, such as image rotation, cannot be directly applied to feature maps generated from fibre optic-sensing data.

In [8], the authors designed a DNN to identify and classify external intrusion signals from a 33 km optical fibre-sensing system in a real environment. In that article, the time-domain data was located directly into a DL model to deeply learn the characteristics of the destructive intrusion events and establish a reference model. This model included two CNN layers, one linear layer, one LSTM layer, and one fully connected layer as shown in Figure 19. It was called the convolutional, long short-term memory, fully connected deep neural network (CLDNN). The model effectively learned the signal characteristics captured by the DAS and was able to process the time-domain signal directly from the distributed optical fibre vibration-monitoring systems. It was found to be simpler and more effective than feature vector extraction through the frequency domain. The experimental results demonstrated an average intrusion event recognition rate exceeding 97% for the proposed model. Figure 20 shows the DAS system using the F-OTDR and the pattern recognition process using the CLDNN. However, the proposed model was not evaluated as a prospective solution for addressing the issue of sample contamination caused by external environmental factors, which can lead to a decline in the recognition accuracy. Other related work can be viewed in [127].

A novel method was developed in [12] to efficiently generate a training dataset using GAN [128]. End-to-end neural networks were used to process data collected using the DAS system. The proposed model’s architecture utilized the VGG16 network [23]. The purpose of the proposed model was to detect and localize seismic events. One extra convolutional layer was added to match the image size then a fully connected layer was added at the end of the model. Batch normalization for regularization and an ReLU activation function were used. The model was tested with experimentally collected data with a 5 km long DAS sensor, and the obtained classification accuracy was 94%. Nevertheless, achieving a reliable automatic classification using the DAS system remains computationally and resource-intensive, primarily due to the demanding task of constructing a comprehensive training database, which involves collecting labelled signals for different phenomena. Furthermore, overly complex approaches may render real-time applications impractical, introducing potential processing-delay issues. Other works in the same direction have been presented in [129,130].

In [127], the authors presented a DL model to recognize six activities, including walking, digging with a shovel, digging with a harrow, digging with a pickaxe, facility noise, and strong wind. The DAS system based on F-OTDR was presented along with novel threat detection, signal conditioning, and threat classification techniques. The CNN architecture used for the classification was trained with real sensor data and consisted of five layers, as illustrated in Figure 21. In this algorithm, an RGB image with dimensions 257 × 125 × 3 was constructed. This image was constructed for each detection point on the optical fibre, helping determine the classification of the event through the network. The results indicated that the accuracy of the threat classification exceeded 93%. However, increasing the depth of the network structure in the proposed model will unavoidably results in a significant slowdown in the training speed and potentially lead to overfitting.

In the study published in [131], the authors proposed an approach to detect defects in large-scale PCBs and measure their copper thickness before the mass production process using a hybrid optical sensor HOS based on CNN. The method involved combining microscopic fringe projection profilometry (MFPP) with lateral shearing digital holographic microscopy (LSDHM) for imaging and defect detection by utilizing an optical microscopic sensor containing minimal components. This allowed for more precise and accurate identification of diverse types of defects on the PCBs. The proposed approach had the potential to significantly improve the quality control process in PCB manufacturing, leading to more efficient and effective production. The researchers’ findings demonstrate a remarkable success rate with an accuracy of 99%.

### 4.2. Multilayer Perceptron (MLP)-Based Applications

In [132], MLP was proposed to achieve a specific event measurement in the existence of various noises without shielding the sensor against undesired perturbations. The proposed model was used for temperature sensing based on a sapphire crystal optical fibre (SOF). MMF interference spectra inclusive were used as the input of temperature changes and noise. The trained DNN was able to recognize the relationship between the temperature and transmission spectra, as shown in Figure 22. The proposed model consisted of four hidden layers. An Adam optimizer with a learning rate of $10^{-3}$ was utilized alongside an ReLU activation function for each output. However, due to the restrictions of the demodulation terminal, the demodulation speed was slow, and as a result, it had a limited scope for application.

In [133], MLP and CNN were used to demonstrate DL for improving the analysis of specklegram analysis for sensing air temperature and water immersion lengths. A comparison was made between the CNN and a traditional correlation technique. The input of the MLP was a 60×60 input image fed into 3600 nodes as the input layer. On the other hand, the output layer comprised a single node, representing a value of either temperature or immersion length. An ReLU activation function was also used after each hidden layer. The total number of trainable parameters was 9,752,929. On the other hand, VGG-16 architecture was used for the CNN model with 2014 input images. In the CNN model, the total number of trainable parameters was 29,787,329. The architecture of both models is shown in Figure 23. Both models obtained better accuracy in terms of their average errors.

### 4.3. Autoencoder (AE)-Based Applications

In [134], a novel deep AE model was proposed to detect water-level anomalies and report abnormal situations. Time-series data were collected from various sensors to train the proposed model, consisting of steps that included pre-processing data, training the model, and evaluating the model using normal and abnormal data, as shown in Figure 24. Combinations of hyperparameters were tuned to obtain the best results from the configuration of each experiment. Different architecture models were used (models through models). These model architecture models differed from each other by the number of units at each layer within the five layers. The studies concluded that the model with 600 × 200 × 100 × 200 × 600 achieved the best result with an F1-score of 99.9% and an area under the curve (AUC) of 1.00 when a window size of 36,000 was used.

In [135], a DL model based on a distributed optical fibre sensor (DOFS) was proposed to collect data concerning the temperature data along the optic fibre and identify the anomaly detected temperature in the early phase. The proposed model had the potential to be used for monitoring abnormal temperatures in crude oil tanks. The structured network used is shown in Figure 25 and Table 15, Table 16, Table 17 and Table 18.

The temperature collected by the DOFS was used as the normal temperature (NT) and used as a training set. The threshold value for anomaly detection was set using NT and a small amount of artificially added ambient temperature (AAT). The test set comprised self-heating temperature (ST) and AAT and NT collected from the experimental apparatus. Furthermore, the proposed model achieved an accuracy of 98%.

Table 19 provides a summary of the DL techniques used with optical sensor applications in this article. The CNN was used in applications from 1 to 16, MLP was used in application 17, and AE was used in the remaining applications. The table also shows the applications and their findings in terms of accuracy. However, the most common limitation of all the previous applications includes the methods to collect and pre-process the data. In addition, there are few DL models shown to be appropriate to be used with optical sensors. Moreover, the used methods for identifying anomalies are very simple, focusing solely on the impact of ambient temperature changes on the detection of sulphurized rust and self-heating anomalies. This method did not consider diverse weather conditions, such as intense winds, rainfall, or temperature variations resulting from seasonal changes or daily fluctuations.

## 5. Conclusions and Future Perspectives

This study summarized the applications of DL in integration with optical sensors. The necessity and significance of DL in optical sensor applications were demonstrated first by presenting the merit of DL, and then by presenting some past and present related works summarized and discussed to provide a wide view of the recent development in this field. This study was based on the type of DL model used. It was concluded that the most commonly used models were the CNN, MLP, and AE models due to the fact that they are suitable for most optical sensor applications. It was also noted that the main challenges in combining DL with optical sensor devices concern the type of data used and how it could be collected and pre-processed before feeding it into the DL model. The treatment of an image classification problem using MLP requires converting a 2D image into a 1D vector before training the model. Two key issues are faced with this approach, the number of parameters increases significantly as the image size increases, and the MLP ignorance of the spatial arrangement, or spatial features, of the pixels in an image. For these reasons, it is better to use CNNs to deal with image classification models. Furthermore, since the data of optical sensors can be modelled as 2D arrays or images, CNNs have become dominant models in various optical sensor applications.

Finally, there are some promising applications that can be considered when DL is combined with some optical sensor applications, such as detecting viruses and bacteria, environmental pollutants, smart city, and optical communication systems. In addition, upon our research, it was apparent that there was a noticeable gap in the literature regarding the application of some modern networks, such as graphical neural networks (GNN) and spiking neural networks (SNN) using optical sensors. Therefore, this article brings attention to this gap and highlights the potential for future research in exploring the utilization of GNN and SNN in optical sensor applications.

## Figures and Tables

**Figure 1 sensors-23-06486-f001:**
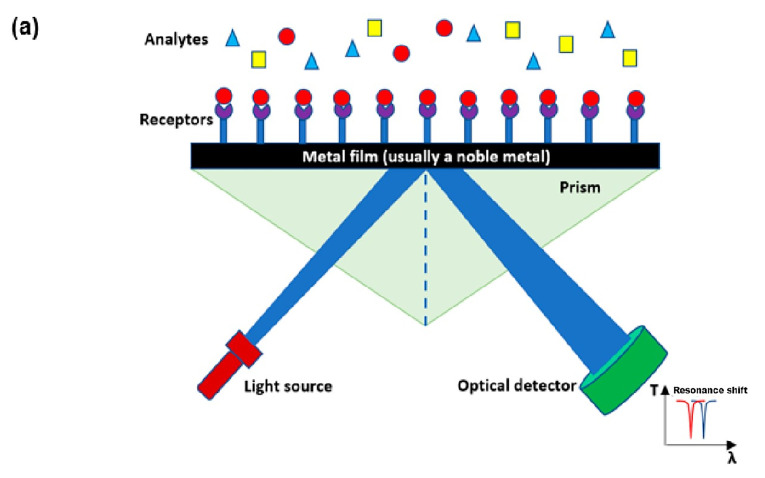
(**a**) A diagram of the mechanism of plasmonic optical sensors, and (**b**) stages of the SPR sensor from detecting analytes to detachment to be reused [36].

**Figure 2 sensors-23-06486-f002:**
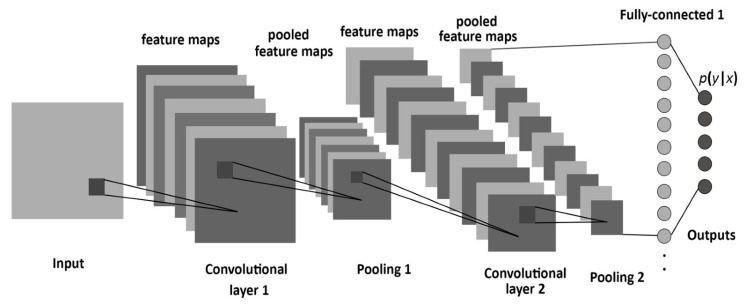
CNN architecture.

**Figure 3 sensors-23-06486-f003:**
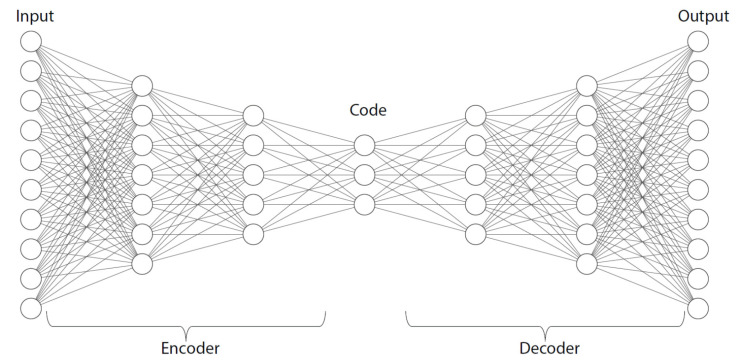
Diagram of the autoencoder.

**Figure 4 sensors-23-06486-f004:**
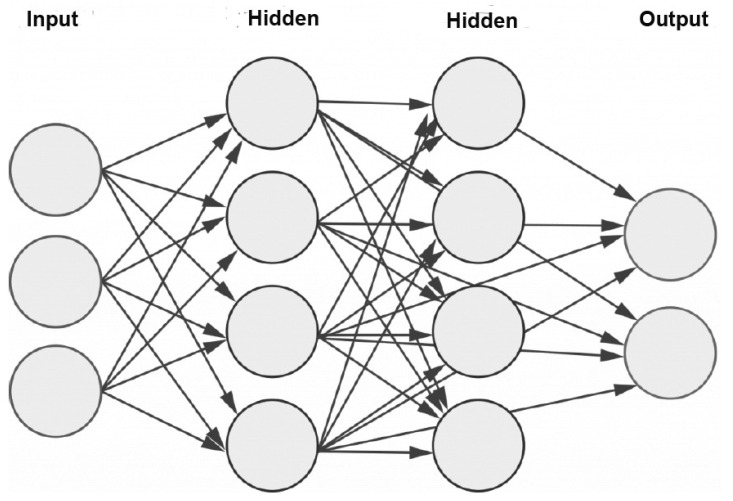
Schematic representation of an MLP with two hidden layers.

**Figure 5 sensors-23-06486-f005:**
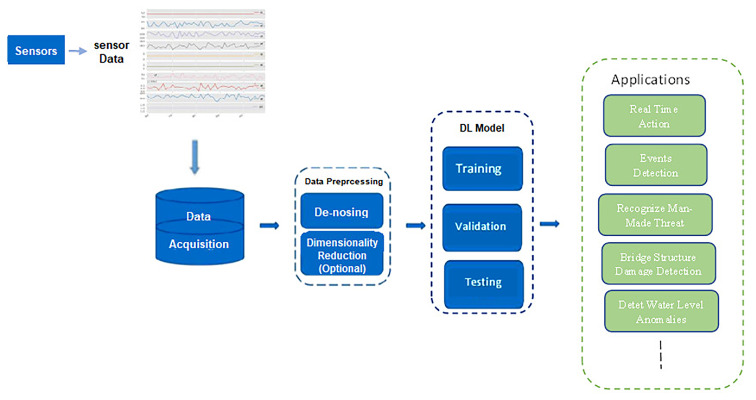
General view of DL with optical sensor applications.

**Figure 6 sensors-23-06486-f006:**
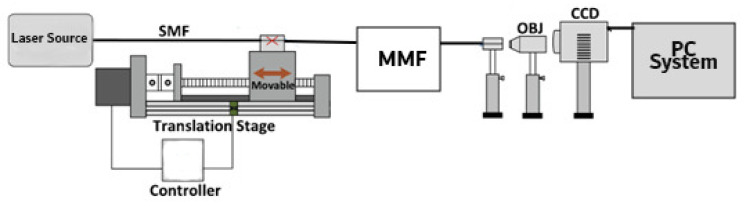
Experimental setup for the detection of fibre specklegrams under different curvatures.

**Figure 7 sensors-23-06486-f007:**
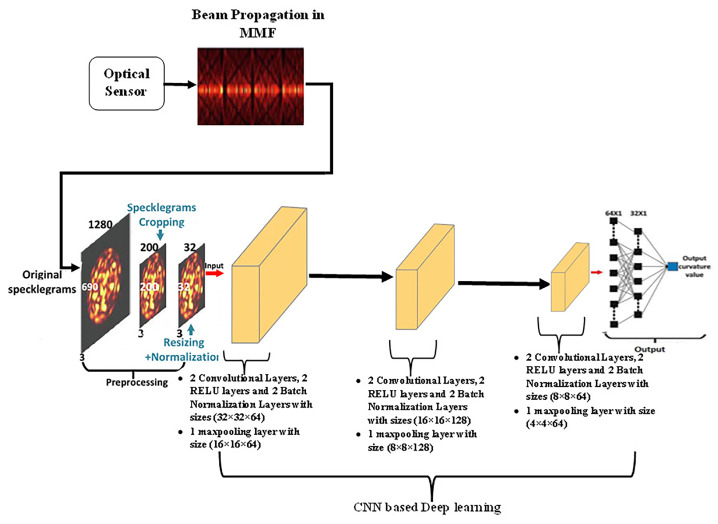
The adopted VGG-Nets architecture to build the CCN.

**Figure 8 sensors-23-06486-f008:**
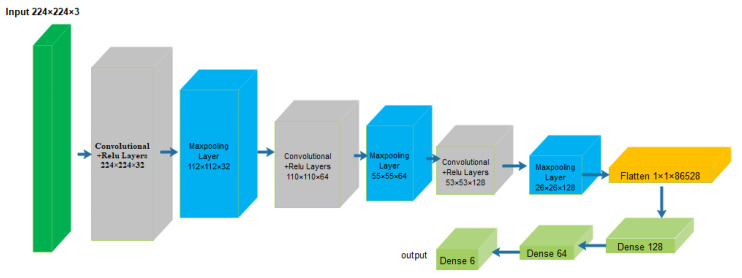
Structural diagram of the CNN used in [95].

**Figure 9 sensors-23-06486-f009:**
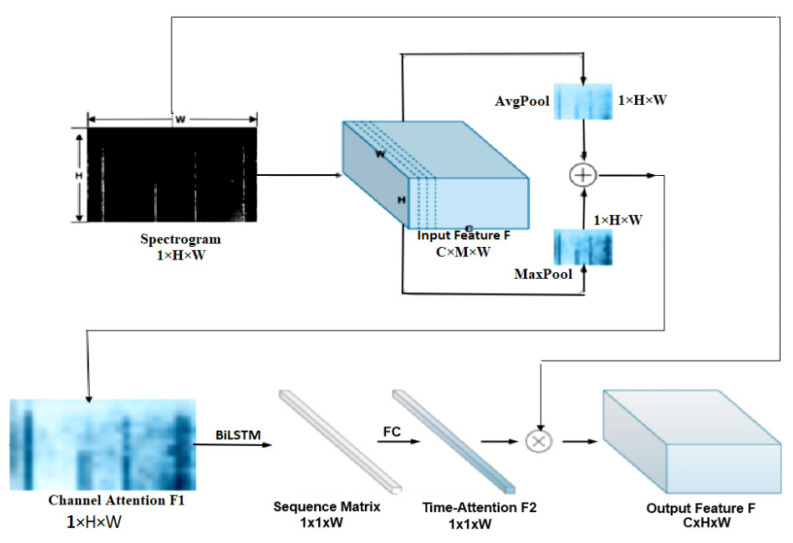
The structure of vibration-sensing system working with F-OTDR.

**Figure 10 sensors-23-06486-f010:**
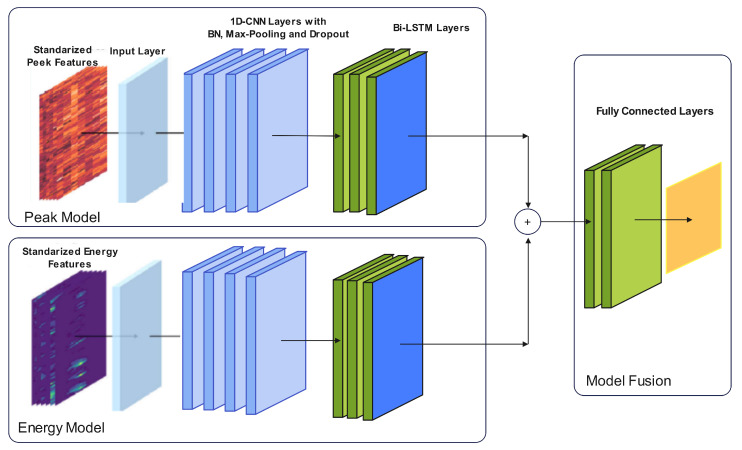
The architecture of the proposed model used in [99].

**Figure 11 sensors-23-06486-f011:**
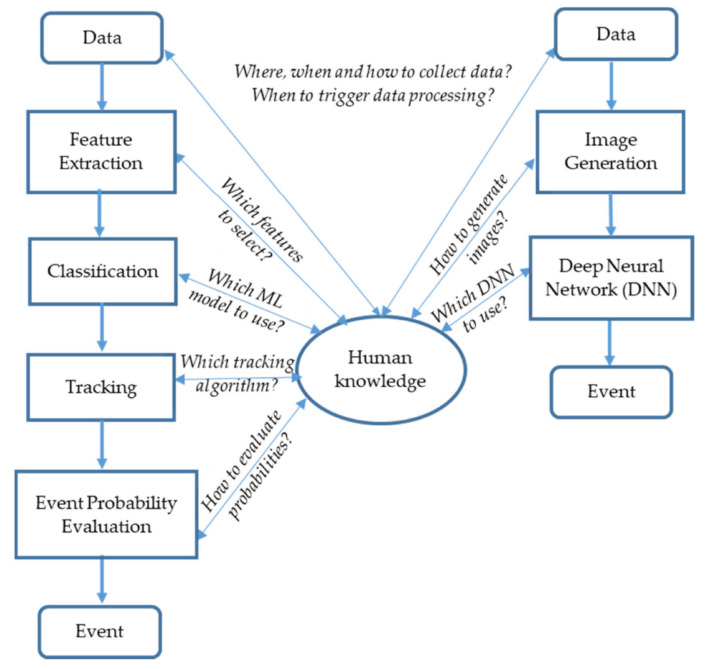
Two methods to detect events by DAS: classic ML approach (left) and DNN approach (right). Note the role of human knowledge [103].

**Figure 12 sensors-23-06486-f012:**
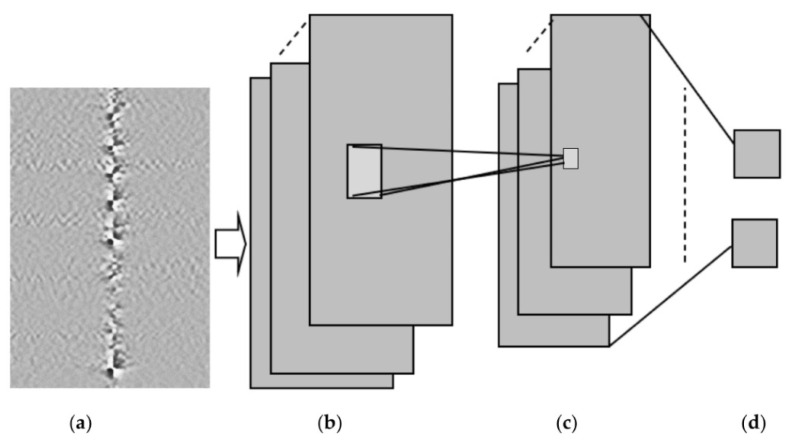
CNN to detect an excavator. (**a**) Input image, (**b**) convolutional layer, (**c**) max pooling layer, and (**d**) fully connected layer [103].

**Figure 13 sensors-23-06486-f013:**
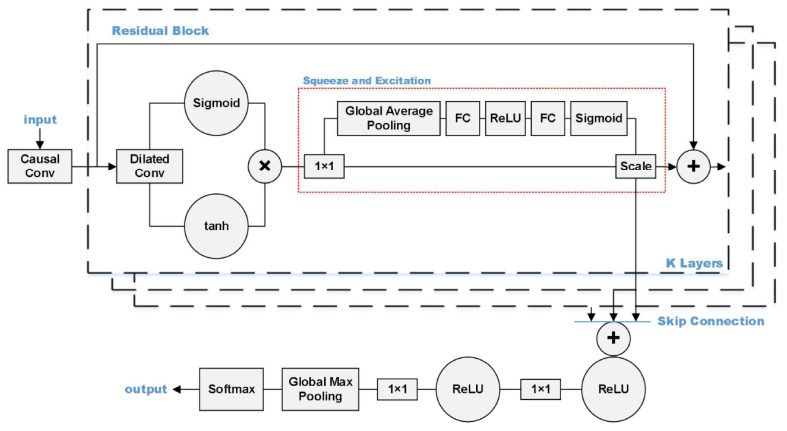
The structure of the model proposed in [104].

**Figure 14 sensors-23-06486-f014:**
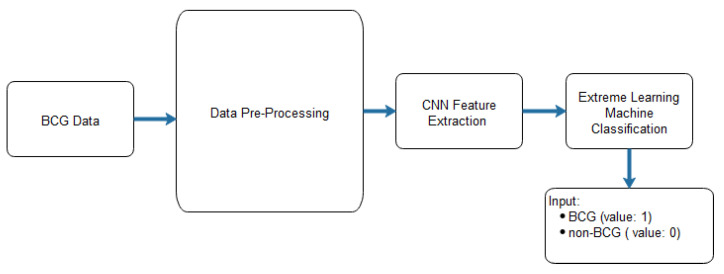
The CNN architecture proposed in [14].

**Figure 15 sensors-23-06486-f015:**
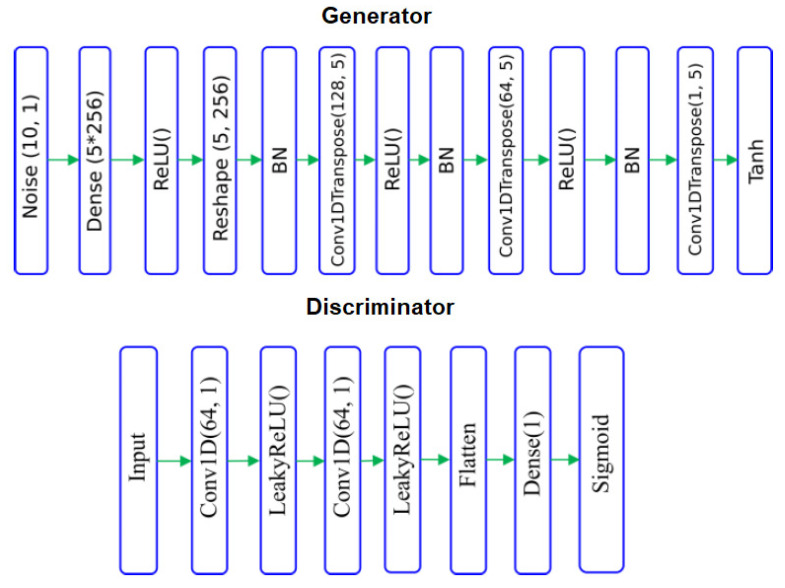
The model architecture of the BCG and non-BCG chunks.

**Figure 16 sensors-23-06486-f016:**
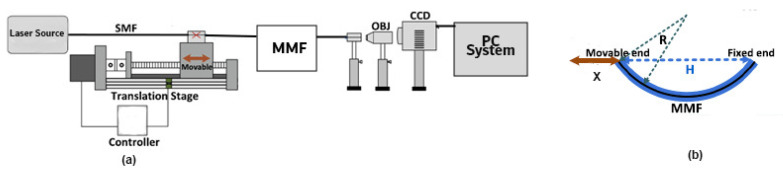
The experimental schematic setup. (**a**) Fibre specklegram detection. (**b**) A graphical depiction of the moving distance x and bending radius R of the translation stage.

**Figure 17 sensors-23-06486-f017:**
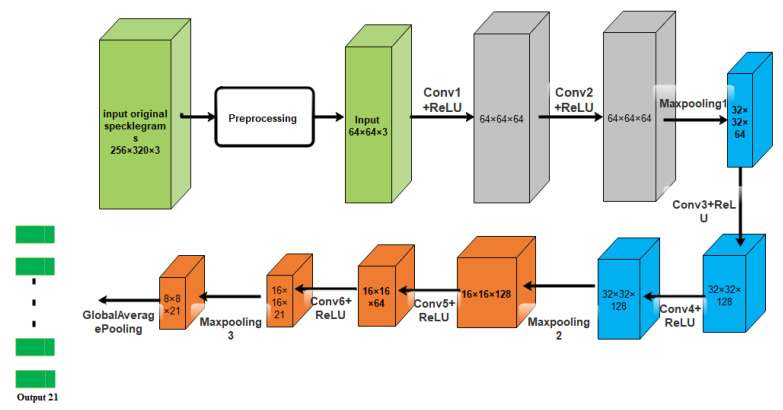
The model structure of the CNN proposed in [113].

**Figure 18 sensors-23-06486-f018:**
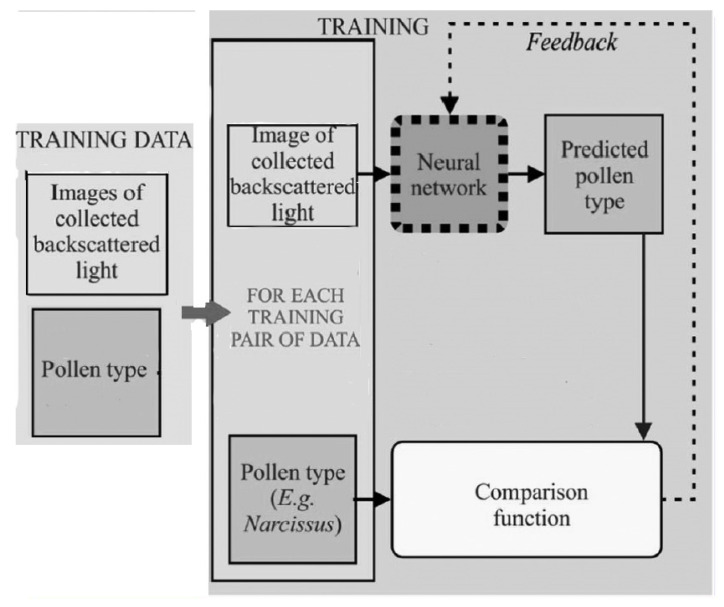
The training procedure of the model proposed in [122].

**Figure 19 sensors-23-06486-f019:**
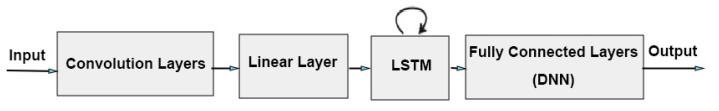
The CLDNN architecture.

**Figure 20 sensors-23-06486-f020:**
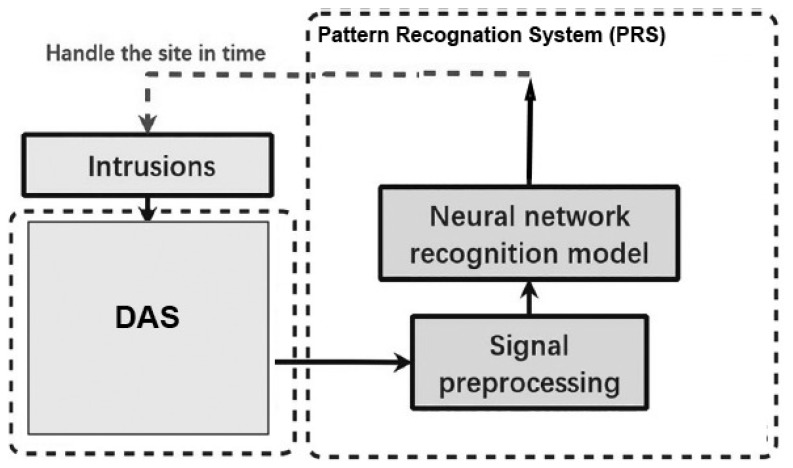
The architecture of the intelligent alarm system proposed in [8].

**Figure 21 sensors-23-06486-f021:**
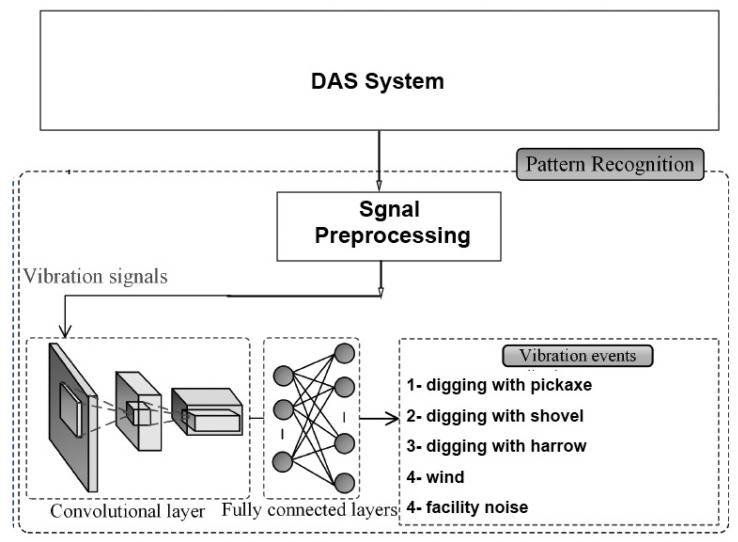
Structure of an F-OTDR and a CNN used for threat classification in [127].

**Figure 22 sensors-23-06486-f022:**
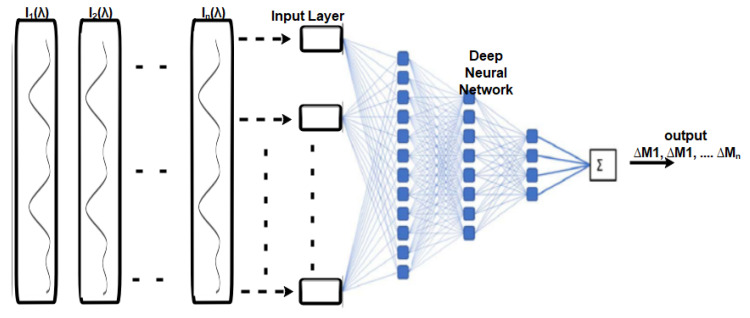
The multilayer perceptron architecture proposed in [132].

**Figure 23 sensors-23-06486-f023:**
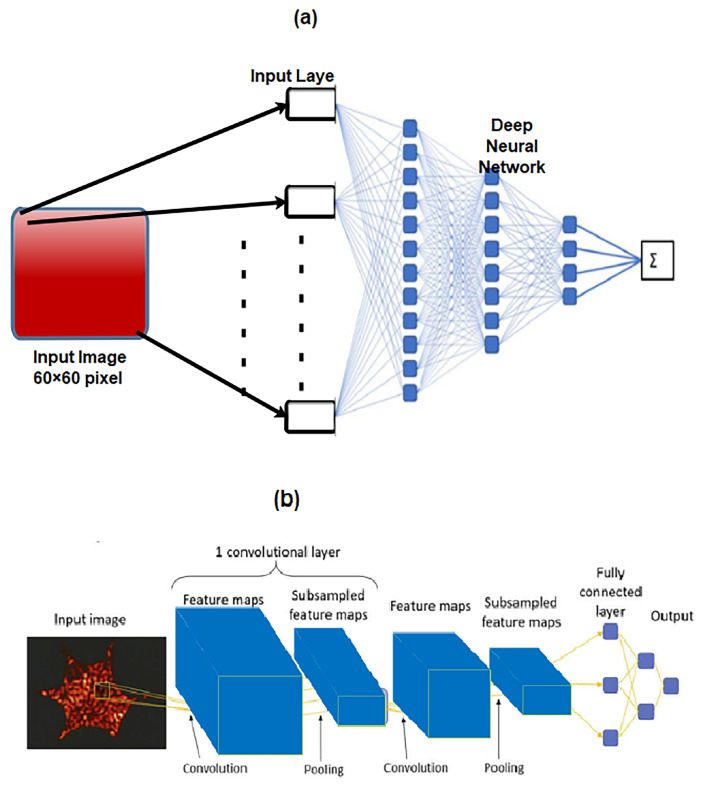
(**a**) Schematic of the DNN. (**b**) Schematic representation of the CNN used in [133].

**Figure 24 sensors-23-06486-f024:**
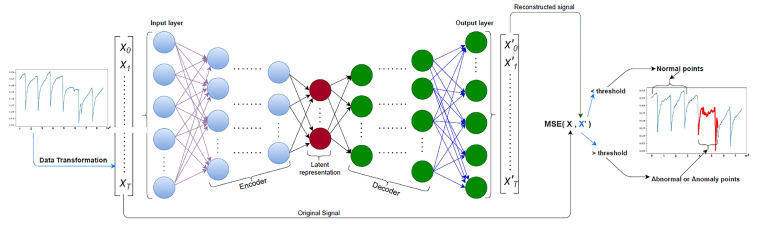
AEs used for the detection of abnormal temperature changes.

**Figure 25 sensors-23-06486-f025:**
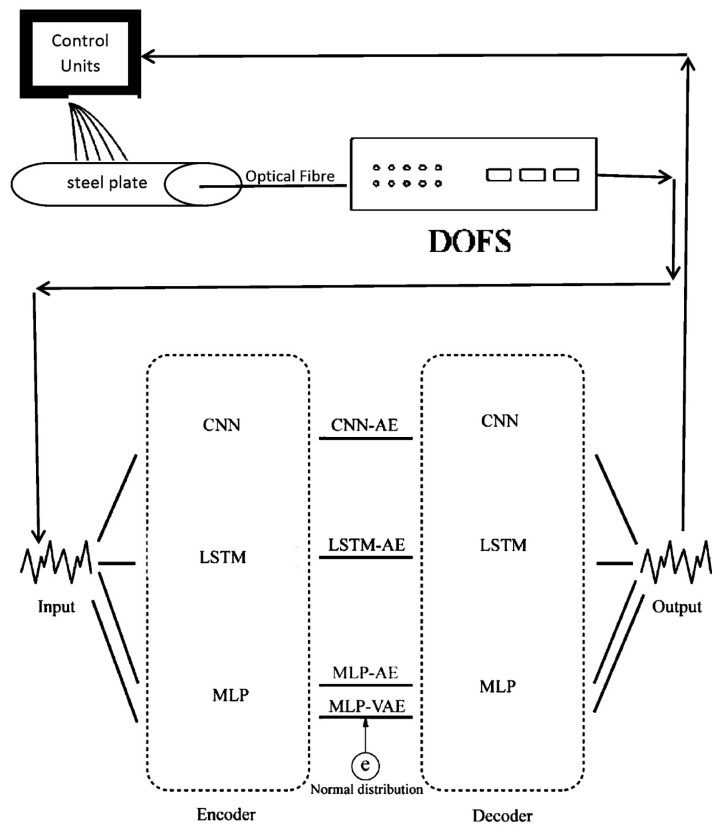
A pipeline for anomaly detection using a DNN model based on AEs in [135].

**Table 1 sensors-23-06486-t001:** Events distribution.

Event	Location
Crevice	120 m, 790 m, 830 m, 1010 m, 1270 m
Beam Gap	400 m, 500 m, 600 m, 700 m, 800 m, 900 m, 1000 m, 1100 m, 1200 m, 1300 m, 1400 m
Cracking	100 m, 480 m, 730 m, 1030 m, 1420 m
Bulge	420 m, 560 m, 730 m, 1030 m, 1420 m
Switches	200 m, 350 m, 450 m, 1350 m
Highway Below	300 m, 750 m, 1250 m

**Table 2 sensors-23-06486-t002:** Dataset obtained after augmentation and the training dataset.

Size	Channels	Event	Dataset	Training Data
5×6	3	Crevice	6853	2708
5×6	3	Beam gap	6853	2788
5×6	3	Cracking	6853	2735
5×6	3	Bulge	6853	2558
5×6	3	Switches	6853	2776
5×6	3	Highway below	6853	2736
5×6	3	No event	6853	2716

**Table 3 sensors-23-06486-t003:** Structural hyperparameters.

Hyperparameters	Options
Structure of the deep learning network	VGG-16, ResNet, Inception-v3, AlexNet, Mobilenet-v3, LeNet
Data balance method	SMOTE-TL, S-ENN, Border-1, MWMOTE, Safe-level
SSL model	Fix-match, Tri-training, UDA
Time interval (t)	[0, 54]

**Table 4 sensors-23-06486-t004:** Dataset details.

Type	Amount
Cutting	948
Impacting	737
Knocking	1235
Rocking	550
Trampling	849
Wind	1169
Total	5488

**Table 5 sensors-23-06486-t005:** Performance comparison of the CNN and MLP algorithms.

ML Algorithm	Accuracy 99%	Exec. Time (µs)
MLP + feature	extraction	99.88% 554.63
CNN	99.91%	34.33

**Table 6 sensors-23-06486-t006:** The number of each event in dataset.

Event Type	Climbing	Violent Demolition Net	Digging	Electric Drill Damage	Total Number
Training set	4319	3616	3713	3825	17,564
Validation set	539	542	562	549	2192
Testing set	539	542	562	549	2192
Total number	5397	5428	5625	5498	21,948

**Table 7 sensors-23-06486-t007:** The proposed CNN architecture.

Event Type	Climbing	Violent Demolition Net	Digging	Electric Drill Damage	Total Number
Layer	No of filters	Activation function	Kernel size	Strides	Output size
Input					(50, 1)
Conv1D	50	ReLU	5	1	(46, 40)
Maxpooling1D		Max	2	2	(23, 50)
Conv1D	50	ReLU	4	1	(20, 50)
Maxpooling1D		Max	2	2	(10, 50)
Conv1D	50	ReLU	4	1	(7, 50)
Maxpooling1D		Max	2	2	(3, 50)
Flatten					(None, 150)
FC1		ReLU			(None, 50)
FC2		Softmax			(None, 2)

**Table 8 sensors-23-06486-t008:** Results of the proposed CNN-ELM.

Data-balancing method	Undersampling	Accuracy	0.89
		Precision	0.9
		Recall	0.87
		F-score	0.88
	Oversampling	Accuracy	0.88
		Precision	0.93
		Recall	0.81
		F-score	0.87
	GAN	Accuracy	0.94
		Precision	0.9
		Recall	0.98
		F-score	0.94

**Table 9 sensors-23-06486-t009:** Hyperparameters.

Bath Size	Epoch	Patience in Early Stopping	Adam			
			α	$\beta_{1}$	$\beta_{2}$	ϵ
128	5000	500	0.001	09	0.009	$1.0\times 10^{-8}$

**Table 10 sensors-23-06486-t010:** Results of the average accuracy.

Fibre	Number of Training Data	Number of Testing Data	Average Accuracy
105	6300	2100	92.8%
200	6300	2100	96.6%

**Table 11 sensors-23-06486-t011:** The number of each data type.

Event Type	I	II	III	IV	V
Training Set	307	1122	1101	1237	748
Validation Set	77	280	275	310	187
Total Number	384	1402	1376	1547	935

**Table 12 sensors-23-06486-t012:** The common CNNs performance.

Model Name Training	Model Size (MB)	Speed (step/s)	Classification Accuracy (%)	Top 2 (%)
LeNet	39.3	90.9	60	86.5
AlexNet	554.7	19.6	94.25	99.08
VggNet	1638.4	2.53	95.25	100
GoogLeNet	292.2	4.1	97.08	99.25
ResNet	282.4	7.35	91.9	97.75

**Table 13 sensors-23-06486-t013:** The classification accuracy achieved for the five events.

Type of Accuracy	I	II	III	IV	V
Accuracy (%)	98.02	98.67	100	92.1	95.5
Top 2 Accuracy (%)	100	100	100	99	100

**Table 14 sensors-23-06486-t014:** The comparison between the optimized model and the Inception-v3.

Network	Accuracy %	Top 3 Accuracy %	Training Speed	Model Size (MB)
The Optimized Network	96.67	99.75	35.61	20
Inception-v3	97.08	99.25	4.35	292.2

**Table 15 sensors-23-06486-t015:** Structures of the MLP-AE.

Composition	Type	Input Size	Output Size
Encoder	Linear	140	32
	Linear	32	4
Decoder	Linear	4	32
	Linear	32	140

**Table 16 sensors-23-06486-t016:** Structures of the MLP-VAE.

Composition	Type	Input Size	Output Size
Encoder	Linear	140	64
	Linear Reparametrization	64	4
Decoder	Linear	4	64
	Linear	64	140

**Table 17 sensors-23-06486-t017:** Structures of the LSTM-AE.

Composition	Type	Input	Output
Encoder	LSTM Repeat	1	32
Decoder	LSTM	32	32
	Dense	32	1

**Table 18 sensors-23-06486-t018:** Structures of the CNN-AE.

Composition	Type	Input Channels	Output Channels	Kernel Size/Stride/Padding
Encoder	Conv.	1	16	5/2/1
	ReLU			
	Maxpool			2/2/0
	Conv.		16	8 5/2/1
	ReLU			
	Maxpool			2/2/0
	Conv.		8	2 3/1/1
	ReLU			
	Maxpool			2/2/0
Decoder	Conv-Transpose	2	16	5/4/0
	ReLU			
	Conv-Transpose	16	8	5/4/0
	ReLU			
	Conv-Transpose	8	1	3/2/1

**Table 19 sensors-23-06486-t019:** A summary of the surveyed DL for optical sensor applications.

No	Applications	Finding	References
1	Realizing an optical fibre curvature sensor. A large number of specklegrams were detected from the facet of a multimode fibre (MMF) automatically in the experiments.	94.7%	[86]
2	Detection of a high-speed railway track	97.91%	[9]
3	Solving the problems of the inability of a conventional hybrid structure to locate in the near field and flawed frequency responses.	97.83%	[95]
4	Extracting the correlation of a time–frequency sequence from signals and spectrograms to improve the robustness of the recognition system.	96.02%	[96]
5	Describing the situation of long-distance oil–gas PSEW systems.	99.26% and 97.20%	[99]
6	Event detection.	99.91%	[103]
7	Discriminate between ballistocardiogram (BCG) and non-BCG signals.	94%, 90%, 98%, and 94%	[14]
8	Recognize a man-made threat event	97.73%	[104]
9	Bridge structure damage detection.	96.9%	[11]
10	An intrusion pattern recognition model	97.67%	[111]
11	Bending recognition model using the analysis of MMF specklegrams with diameters of 105 and 200 μm.	92.8% and 96.6%	[113]
12	Demonstrate the capability for the identification of specific species of pollen from backscattered light.	97%	[122]
13	Event recognition.	99.75%	[13]
14	Identification and classification of external intrusion signals in a real environment using a 33 km optical fibre-sensing system.	97%	[8]
15	Detection and localization of seismic events.	94%	[12]
16	Identification of six activities, walking, digging with a shovel, digging with a harrow, digging with a pickaxe, strong wind and facility noise.	93%	[127]
17	Temperature sensing based on a sapphire crystal optical fibre (SOF).	99%	[132]
18	Sensing water immersion length measurements and air temperature.	N/A	[133]
19	Detecting water-level anomalies and reporting abnormal situation.	99.9%	[134]
20	Collection of temperature data along an optic fibre set and identify the anomaly detected temperature in the early phase.	98%	[135]
21	Detection of defects on large-scale PCBs and measure their copper thickness before the mass production process	[131]	

## Data Availability

The data in this study are available from the corresponding author upon request.

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
