# Peer review of "Deep Learning for Optical Sensor Applications: A Review"

_sensors, 2023, doi:10.3390/s23146486_

Round 1

Reviewer 1 Report (Previous Reviewer 1)

I think the paper is improved to be acceptable

Reviewer 2 Report (Previous Reviewer 2)

Dear Authors, 

on the basis of the answers i suggest to accept the paper

Moderate English editing

This manuscript is a resubmission of an earlier submission. The following is a list of the peer review reports and author responses from that submission.

Round 1

Reviewer 1 Report

1. The quality of the images in the article is insufficient. Such as Figure 1 and Figure 14 with low resolution, Figure 2 and Figure 6 with uncomfortable font scale.

2. The content in Section 3 is too basic for deep learning. It doesn't explain the deep learning in optical sensors application. Therefore, there is no need for this section to exist.

3. The content of Section 4 is a simple list of existing research, without analyzing its internal development patterns and future development directions. For example, curvature prediction, vibration classification, etc are essentially using CNN for classification, and there is no need to introduce so many types of application. On the contrary, there is a lack of relatively new architectures such as GNN and SNN in this type of sensor application, the author needs to analyze whether these new architectures to the same data types can be applied to such sensors.

Reviewer 2 Report

Dear Authors,

Major revisions are needed prior to manuscript acceptance

The quality of English is adequate, for this reason, moderate improvement is requested